# Association of Stress and Inflammatory Diseases with Serum Ferritin and Iron Concentrations in Neonatal Calves

**DOI:** 10.3390/ani15071021

**Published:** 2025-04-02

**Authors:** Marlene Sickinger, Jessica Jörling, Kathrin Büttner, Joachim Roth, Axel Wehrend

**Affiliations:** 1Clinic for Ruminants and Herd Health Management, Justus-Liebig-University of Giessen, 35392 Giessen, Germany; 2Veterinary Practice Lünne, 48480 Lünne, Germany; jessica.joerling1@gmx.de; 3Department for Biomathematics and Data Processing, Justus-Liebig-University of Giessen, 35392 Giessen, Germany; kathrin.buettner@vetmed.uni-giessen.de; 4Institute for Veterinary Physiology, Justus-Liebig-University of Giessen, 35390 Giessen, Germany; joachim.roth@vetmed.uni-giessen.de; 5Veterinary Clinic for Reproduction and Neonatology, Justus-Liebig-University of Giessen, 35392 Giessen, Germany; axel.wehrend@vetmed.uni-giessen.de

**Keywords:** iron supply, inflammation, cortisol, cytokines, hepcidin, haptoglobin, calf

## Abstract

Iron metabolism in organisms is influenced by various factors, including nutrition, inflammatory diseases, and immune status. Neonatal calves can develop iron deficiency anemia and inflammatory conditions, such as diarrhea and omphalitis. Iron supplementation is commonly administered within the first days of life to prevent anemia. However, the relationship between serum iron and ferritin concentrations and inflammatory markers depending on iron supplementation and health status in early life remains poorly understood. Therefore, we investigated stress- and inflammation-associated parameters, including cortisol, white blood cell count, total protein, lactate, interleukin 1β, interleukin 6, substance P, hepcidin, haptoglobin, and the ferric-reducing ability of plasma, in neonatal calves over the first 10 days of life in relation to serum iron and ferritin concentrations. Additionally, we examined the effects of iron supplementation, inflammatory diseases, and potential sex-associated differences on the aforementioned parameters.

## 1. Introduction

Iron is an essential micronutrient, and its deficiency can have a wide range of adverse effects on animals. Calves with iron deficiency exhibit stunted growth and an increased susceptibility to infectious diseases [1,2]. Iron plays a critical role in erythropoiesis and various metabolic processes in the body [3]. Accordingly, it is associated with various inflammatory conditions [4,5]. Iron metabolism is regulated by proteins, such as hepcidin, or indirectly by cytokines, thereby highlighting the hormone-like regulation of iron metabolism [6]. Maintaining iron homeostasis is particularly challenging due to its toxic and proinflammatory nature. Apart from its physiological function in non-enzymatic proteins, such as hemoglobin or myoglobin, and enzymes like cytochromes, the pathogenic potential of this essential trace element lies in its redox potential, which facilitates the formation of reactive oxygen species (ROS). Maintaining iron homeostasis is crucial to protecting cells and tissues from ROS [6,7]. Both iron overload and deficiency result in reduced antioxidative power, low antioxidant enzyme activity in erythrocytes, and diminished total antioxidant capacity in the plasma [7,8]. The ferric-reducing ability of plasma (FRAP) is a simple method to assess this antioxidant capacity [9].

Iron is primarily obtained through dietary intake and water consumption. While roughage and silage are rich in iron, colostrum and milk contain relatively low iron concentrations, which can be further affected by processing and environmental factors [6,10,11]. In addition to the low iron content in colostrum and milk, these sources contain iron-binding proteins such as lactoferrin in varying amounts. Lactoferrin serves as a non-enzymatic antioxidant and reduces cellular inflammation and iron availability for bacterial agents that challenge the immune system of newborn calves [12,13,14]. The preference for white veal has historically led to iron deficiency in calves [15]. Prolonged feeding of calves exclusively with milk, without dietary supplements or solid feed, can result in iron deficiency anemia and abomasal ulcers [16,17]. To counteract this deficiency, calves are commonly treated with injectable iron solutions within the first week of birth [18].

Monitoring serum iron concentrations to diagnose iron deficiency in calves has limitations and a high degree of bias due to the influence of homeostasis, nutrition, iron-binding substances, antagonists, and inflammation [19]. Instead, ferritin measurement is the standard procedure for evaluating iron storage reserves and confirming iron deficiency [20,21,22]. Ferritin serves as the main intra- and extracellular storage protein for iron [6]. Several cells, like hepatocytes, macrophages, and Kupffer cells, are capable of secreting ferritin, which is not only able to store iron but also make it available for cells [23]. Serum ferritin concentrations are correlated to the intracellular iron contents, thus, representing a measure of the iron storage of the organism [21]. The sensitivity of serum ferritin concentrations to iron depletion has been recognized since the 1980s [24]. Periparturient cows exhibit elevated ferritin concentrations due to inflammatory responses associated with delivery [25]. In humans, ferritin thresholds for diagnosing iron deficiency during pregnancy are <12 and <15 ng/mL [26]. However, healthy cattle generally exhibit lower ferritin concentrations than humans [25], thereby suggesting lower cutoff values for diagnosing iron deficiency. Since both serum iron and ferritin concentrations are influenced by inflammatory diseases and elevated proinflammatory cytokine concentrations [25,27], these disturbances complicate diagnostics in affected calves. Proinflammatory cytokines, like IL-1β, IL6, and TNFα, lead to an enhanced secretion of ferritin from hepatocytes and an enhanced expression of hepcidin. Thus, iron absorption is reduced and the relative ratio of ferritin to iron is modulated in animals affected by inflammatory diseases [28]. However, serum ferritin itself is also regarded as an acute phase protein that is nonspecifically enhanced during inflammatory conditions and represents elevated iron storages in such cases. In such cases, the stored iron is sequestered and is not available for hematopoiesis being the underlying correlate for the so-called anemia of inflammation [23].

The aim of this study was to examine stress- and inflammation-associated parameters, such as cortisol, WBC counts, interleukins, hepcidin, haptoglobin, and substance P, in neonatal calves in relation to serum iron and ferritin concentrations. The effects of iron supplementation via injection and potential sex-associated differences on the aforementioned parameters were also investigated.

## 2. Materials and Methods

The general study protocol has been published previously [29]. Below, we provide a detailed description of the current phase of this study.

### 2.1. Animals, Housing, and Feeding

All examinations and treatments conducted in this study were approved by the Giessen Regional Council (V 54–19 c 20 15 h 01 GI 18/14; Nr. G 79/2019) and adhered to the ARRIVE 2.0 guidelines. This study was conducted between February 2020 and September 2021 and involved 40 neonatal German Holstein calves (20 males and 20 females). The calves were acquired promptly after birth from nearby dairy farms situated within a distance of less than 15 km from Justus-Liebig University in Giessen, Germany.

Immediately after birth, the calves underwent a clinical examination, were separated from their dams, and were given an initial colostrum supply of 2–3 L from their dams on the first day. The amount of colostrum supply per feeding depended on the body weight of the calves, with at least 6 L of colostrum fed during the first 12 h of life. During the 10 d study period, the calves were housed individually in straw-bedded boxes within the facilities of the Clinic for Ruminants, JLU Giessen. After finishing the study period, they were moved to group pens with straw bedding. Temperature and humidity depended on the surrounding climate, and ventilation consisted of passive ventilation with exhaust air.

A second colostrum feeding was administered within the first 6 h after birth, followed by three daily feedings of a commercial milk replacer (CombiMilk Start, Agravis Raiffeisen AG, Münster, Germany; Fe(II): 150 mg/kg DM). From the second day after birth, hay, concentrate feed (Kälberkraft; Agravis Raiffeisen AG, Münster, Germany), and water were provided ad libitum.

### 2.2. Experimental Procedures

At their farms of origin, the calves underwent a comprehensive clinical examination to confirm their suitability for this study. This assessment included evaluating key health indicators, such as physical appearance, heart rate, respiration, and suckling reflex, along with screening for congenital malformations or a shortened, thickened navel. The animals’ navels were treated with iodine shortly after birth as a standard procedure. The initial blood sample was obtained through jugular venipuncture using an 18 G cannula. Upon relocation to the clinic, the animals were randomly assigned to either the treatment or control groups. The treatment group received 10 mg/kg of iron dextran intramuscularly (belfer, bela-pharm, GmbH & Co. KG, Vechta, Germany) on the first day. This dosage represents the standard dosage recommended by the manufacturer and is in line with the standard protocols used in the field within the catchment area of the clinic. The control group did not receive any mineral, trace element, or vitamin supplements. During the 10 d study, daily clinical examinations were performed, which included evaluating their general health status, monitoring vital signs, palpating the umbilical region and limb joints, and a macroscopical analysis of the feces.

From days 2 to 10, blood samples were collected via a jugular vein catheter (PUR Infusionskatheter^®^, 16 G, soft tip, 15 cm; Walter Veterinär-Instrumente e.K., Baruth, Germany) inserted on the second day after birth. The jugular vein was examined twice daily for signs of inflammation. Blood samples were collected in serum-separating tubes containing ethylenediaminetetraacetic acid (EDTA) and lithium heparin (Kabe, Nümbrecht, Germany). For serum separation, the samples were allowed to clot for 30 min before centrifugation (20 min, 21 °C, 1000× *g*). The separated serum was then stored at −80 °C until analysis.

### 2.3. Blood Parameters

WBC counts (reference range, 5.6–14.3 G/L) were determined using the IDEXX ProCyte Dx cell counter (IDEXX Laboratories, Kornwestheim, Germany). Serum iron concentrations (reference range 14.5–25 mmol/L, Test kit LT-SI 0100; Labor und Technik Lehmann GmbH, Berlin, Germany) were measured photometrically at the specific wavelength recommended by the manufacturer. Concentrations of ferritin, IL1β, IL6, hepcidin, and SP were assessed using bovine ELISA plates (Cloud-Clone Corp., Katy, TX, USA; intra- and inter-assay coefficients of variation: <10% and <12%). Haptoglobin analyses were conducted externally using a commercial test kit (PHASE Haptoglobin Assay Cat. No. TP-801; TriDelta Development Ltd., Kildare, Ireland; intra- and inter-assay coefficients of variation: 6.3% and 4.1%, detection limit: 0.005 mg/mL.). FRAP concentrations were determined using a colorimetric assay kit (Cell Biolabs Inc., San Diego, CA, USA). Cortisol concentrations were measured using an in-house radioimmunoassay (RIA) test (Endocrine Laboratory of the Veterinary Clinic for Reproduction and Neonatology). Prior to RIA, samples were pretreated using the method of Donohue et al. [30]. Briefly, the samples were diluted to 1:10 in a glutamic acid buffer of pH 3.3 and incubated in a boiling water bath for 30 min. The processed samples were analyzed using an in-house RIA with a tritium-labeled tracer and charcoal adsorption to separate free and antibody-bound hormones. The antiserum was specific to cortisol-3-carboxymethyl oxime-BSA with cross-reactivity to cortisol (100%), corticosterone (3.8%), androstenedione, 5α-dihydrotestosterone, estradiol-17β, pregnenolone, progesterone, and testosterone (<0.01%). The intra- and inter-assay coefficients of variation were 4.3% and 9.8%, respectively, with a detection limit of 2 nmol/L.

### 2.4. Evaluation of Health Status

All calves underwent a comprehensive clinical examination within the first 1–2 h of birth to confirm their suitability for this study (e.g., clinical health, including appearance, heart rate, respiration, and suckling reflex; absence of congenital malformations (i.e., palatine cleft or atresia ani); and intact navel). Throughout the study period, the calves received daily clinical evaluations and scoring of findings to observe their behavior, heart and respiratory rates, core body temperature, presence of diarrhea or pneumonia, and potential inflammatory responses to the indwelling jugular vein catheter or navel condition. Based on clinical findings and scoring, calves were classified as healthy or exhibiting mild, moderate, or severe disease symptoms. Disease severity was determined by the duration of symptomatic days: healthy (0–2 d), mild (2–5 d), moderate (5–7 d), or severe (7–10 d). None of the calves developed pneumonia.

In cases of diarrhea, a parasitological examination of fecal smears stained with carbol fuchsin was performed to detect cryptosporidiosis. Additional parasitological, microbiological, or virological tests were not conducted due to ongoing herd health management measures at the source farms. However, the risk associated with pathogenic bacteria or viral agents was considered minimal.

### 2.5. Statistics and Data Analyses

The Department for Biomathematics and Data Processing at the University of Giessen conducted power and statistical analyses using BiAS Version 9 (Epsilon Publishing, Hochheim, Darmstadt, Germany), and SAS^®^ 9.4 software (SAS^®^ Institute Inc., 2013. Base SAS^®^ 9.4 Procedures Guide: Statistical Procedures, 2nd ed. Statistical Analysis System Institute Inc., Cary, NC, USA), respectively. The sample size was based on a power analysis yielding a sample size of 36 calves (α = 0.05; β = 0.1; biological relevant difference = 1.4 SD). The areas under the curve (AUC) for ferritin, Fe, IL-1β, IL-6, SP, hepcidin, and FRAP were calculated and log-transformed to achieve a normal distribution. These values were compared using a three-way analysis of variance to assess the potential effects of iron treatment, disease status, and sex. The Bonferroni adjustment was applied to account for multiple comparisons. Time-dependent changes in all parameters over the 10-day examination period were graphically represented and analyzed after categorizing the animals based on treatment or sex.

To assess the potential impact of transportation, changes in cortisol concentrations between days 1 and 2 were compared using the least squares means procedure.

Spearman correlation analyses were performed to investigate potential correlations among the parameters. Statistical significance was defined at *p* < 0.05.

## 3. Results

During the experimental period, 32 of the 40 calves exhibited mild-to-severe symptoms, including diarrhea, omphalitis, and/or thrombophlebitis. The diarrhea, which persisted for 1–8 days, was attributed to cryptosporidiosis, and linked to a herd health issue at the farm of origin. All calves recovered either spontaneously (*n* = 3), with oral rehydration therapy (*n* = 20) or intravenous fluid therapy (*n* = 9). No antibiotic treatments were administered during the observation period. Nonsteroidal anti-inflammatory drugs (0.5 mg Meloxicam/kg, Melosolute, CP-Pharma, Burgdorf, Germany; or 40 mg Metamizol/kg, Metamizol WDT, WDT, Garbsen, Germany) were administered to 12 of 40 calves.

Diarrhea was first detected on day 2 (*n* = 6), with mild-to-severe cases observed in 13 calves from day 3 onward and in 11 calves from day 6 onward. Thickening of the jugular vein or an increase in body temperature was noted in two calves from day 3 onward and in five calves from day 5 onward.

Neither iron supplementation, disease status, nor sex had statistically significant effects on the AUC of ferritin, WBC, total protein (TP), interleukin 1β (IL1β), interleukin 6 (IL6), substance P (SP), hepcidin, haptoglobin, and FRAP. However, Fe and cortisol concentrations were considerably influenced by disease development.

As reported earlier [29], the Fe values were influenced by the health status of animals (*p* = 0.02) and were higher in healthy calves than in severely diseased calves (*p* = 0.04). Throughout this study, particularly on day 9, the WBC count in diseased calves exceeded that in healthy calves (14.6 ± 4.4 vs. 10.9 ± 2.5; Figure 1A). However, no statistically significant differences in WBC count between the groups or correlations with ferritin concentrations were observed. Similarly, WBC concentrations were unaffected by treatment, disease status, or sex. Cortisol concentrations were also affected (*p* = 0.002), with healthy calves exhibiting lower concentrations than severely diseased calves (*p* = 0.0499; Figure 1E). The concentration of TP increased from day 1 to 2, showing a consistent pattern across all animals. Diseased calves exhibited slightly higher TP concentrations than healthy calves (Figure 1B). An interaction between disease and treatment was observed (*p* = 0.0288), which did not remain significant following the Bonferroni correction. Haptoglobin concentrations remained stable until day 4, after which higher concentrations of haptoglobin were observed in the diseased calves (Figure 1C). However, no significant effects of the treatment, sex, or disease were noted. Correlation analyses for the AUCs of haptoglobin indicated negative correlations with Fe (*p* = 0.0066; ρ = −0.42), hepcidin (*p* < 0.0001; ρ = −0.62), lactate (*p* = 0.0382; ρ = −0.33), and IL1β (in male calves: *p* = 0.0427; ρ = −0.46). Lactate concentrations decreased substantially from day 1 to 2. Treated animals showed higher lactate concentrations (*p* = 0.0039), although the patterns in both healthy and diseased animals were comparable (Figure 1D).

The significance of these correlations varied based on iron supplementation and/or sex (Table 1) (*p* = 0.0024; ρ = −0.64) and female sex (*p* = 0.0021; ρ = −0.65).

Interleukin 1β (IL1β) exhibited distinct patterns depending on iron supplementation. In the treated group, IL1β concentrations increased postnatally, starting at 36.1 ± 24.6 pg/mL on day 2, then declined until day 6, and subsequently peaked on day 9 at 42.2 ± 60.3 pg/mL. In contrast, control calves exhibited inherently higher IL1β concentrations than the supplemented group. Both IL1β and ferritin concentrations were approximately two-fold higher in control calves, peaking on day 2 at 62.4 ± 66.2 pg/mL and 68.17 ± 100.1 ng/mL, respectively. Positive correlations between ferritin and IL1β concentrations were observed in control animals on day 2 (*p* = 0.0369; ρ = 0.47) and day 3 (*p* = 0.0052; ρ = 0.60) (Figure 2). These correlations were sex-specific in male calves, with *p* = 0.048 and ρ = 0.45 on day 2, and *p* = 0.003 and ρ = 0.63 on day 3.

Correlation analysis of the AUC values showed a weak positive correlation between AUC ferritin and AUC IL1β (*p* = 0.0015; ρ = 0.49). This weak correlation persisted when the data were stratified by group or sex. The IL6 progression mirrored that of ferritin, with both reaching peak concentrations on day 2 (ferritin) and 3 (IL6) in the supplemented calves and on day 2 in the control calves. There was a marked difference in IL6 concentrations with 926.4 ± 1702 pg/mL in treated calves and 539.8 ± 518.9 pg/mL in controls (Figure 3 and Figure 4).

There were strong positive correlations between ferritin and IL6 from days 1–3 (*p* = 0.036; *p* = 0.0048; *p* = 0.0032); on day 4, for male calves exclusively (*p* = 0.041; ρ = 0.46); on day 9, for calves without iron supplementation (*p* = 0.048; ρ = 0.45). There was a weak positive correlation between AUC ferritin and AUC IL6 (*p* = 0.0011; ρ = 0.50) that was consistent across genders but only observed in the control group (*p* = 0.0066; ρ = 0.59).

No significant correlations were observed between the iron-regulating enzyme, hepcidin, and ferritin. Negative correlations between hepcidin and serum iron concentrations were observed (Figure 5).

Non-supplemented animals exhibited a mild negative correlation between iron and hepcidin on day 2 (*p* = 0.0175; ρ = −0.52), with significance observed only in male animals (*p* = 0.027; ρ = −0.495). In female animals, a mild positive correlation was observed between iron concentration and hepcidin concentrations on day 3 (*p* = 0.04; ρ = 0.47). In animals receiving iron supplementation, a positive correlation was noted between iron and hepcidin on day 7 (*p* = 0.012; ρ = 0.55) and day 9 (*p* = 0.03; ρ = 0.499). When grouped by sex, significance was only reached in male calves on days 7, 9, and 10 (*p* = 0.03; ρ = 0.49; *p* = 0.049; ρ = 0.45 and *p* = 0.036; ρ = 0.47). Strong correlations were also observed between ferritin and cortisol concentrations (Figure 6).

On day 1, a weak negative correlation existed between ferritin and cortisol with ρ = −0.41 (*p* = 0.0094). This correlation was also evident in calves of the control group (*p* = 0.04; ρ = −0.46) and particularly in male animals (*p* = 0.0009; ρ = −0.68). The cortisol concentrations decreased continuously from day 1 (44.1 ± 12.0 ng/mL) to day 10 (12.5 ± 8.99 ng/mL), reaching baseline concentrations by day 5 (15.6 ± 9.9 ng/mL). A negative correlation in male animals was also observed for the relationship between AUC serum iron and AUC cortisol (*p* = 0.015; ρ = −0.54).

Substance P concentrations decreased postnatally in both groups until day 7, then returned to near initial concentrations. The initial SP concentrations were 469.9 ± 183.3 pg/mL in the treatment group and 436.1 ± 182.1 pg/mL in the control group. Significant negative correlations between ferritin and SP were observed for male calves on day 1 (*p* = 0.03; ρ = −0.49), control group calves on day 3 (*p* = 0.047; ρ = −0.45), and female calves on day 9 (*p* = 0.036; ρ = −0.47).

A weak negative correlation between ferritin and FRAP on day 8 in the control group was observed (Figure 7). Animals without an iron injection exhibited a slight decrease in FRAP corresponding to elevated ferritin concentrations (*p* = 0.0223; ρ = −0.50771).

## 4. Discussion

Bone marrow aspiration is considered to be the gold standard for assessing iron reserves [26]. However, its invasive nature has led to the use of various blood parameters, including serum iron, serum ferritin, transferrin saturation, and reticulocyte hemoglobin content, to diagnose iron deficiency [31]. However, these parameters can be affected by inflammation, nutrition, chronic diseases, or aging [6,28,31,32,33].

Our previous study demonstrated that compromised health status strongly impacts iron concentration in neonatal calves [29]. Nevertheless, there are limited data on the relationships among serum iron, serum ferritin, and inflammatory markers, including WBC count, TP, lactate, cytokines, substance P, haptoglobin, cortisol, and hepcidin concentrations in neonatal calves. To the best of our knowledge, this is the first report on the effects of parenteral iron supplementation, sex, and disease status on these measured parameters.

The sex-specific findings of the present study are regarded to be the result of hormonal influences on the examined parameters. This is in line with negative correlations of serum ferritin with testosterone in boys [34], and estrogen being involved in the expression of ferroportin [35], thus, influencing iron metabolism. Whereas several studies in human medicine emphasize the influence of testosterone and estrogen on iron metabolism, serum ferritin levels, and anti- or proinflammatory cytokines, corresponding data for veterinary medicine are scarce. Nevertheless, sex-specific differences for substance P in calves, for example, have been shown, which emphasizes the necessity for sex-specific considerations [36]. Recent studies in mice and humans indicate that sex chromosomes and gonadal hormones influence the number and functions of immune cells, cytokine signaling pathways, and humoral responses [37]. Due to a differing expression of X-chromosomal and autosomal genes, and the abundant presence of sex steroid hormone receptors in many cell types, sex influences nearly all biological systems [37]. The future establishment of specific reference ranges and cut-off values according to sex are therefore generally recommended.

Previous studies have shown the impact of transportation and inflammatory conditions, such as diarrhea or bronchopneumonia, on cortisol concentrations in neonatal calves [38,39,40]. However, the results are inconclusive. Whereas Cabello [38] and Masmeijer et al. [40] reported elevated cortisol concentrations in diseased calves, Brückmann et al. [39] found higher cortisol concentrations in healthy calves. The current study indicates that cortisol concentrations were highest at birth and decreased over the evaluation period. Healthy calves consistently exhibited lower cortisol concentrations than did the diseased calves. Additionally, iron supplementation led to a slight reduction in cortisol concentrations, although these variances were not statistically significant. Significance might have been reached using higher dosages of iron or a prolonged oral supplementation of iron and selenium as has been shown in human athletes [41] and young bulls transported over long distances [42]. In athletes, a reduction in cortisol was achieved after oral supplementation of iron over a period of 3 weeks. In bulls, the application of slow-release selenium boluses resulted in an elevation of serum iron and a reduction in cortisol. The link between cortisol and iron metabolism is regarded to be an indirect effect of the promotion of inflammation in acute stress (i.e., higher cortisol concentrations in diseased animals). In animals suffering from acute stress, proinflammatory cytokines are enhanced leading to higher concentrations of hepcidin [6], thereby reducing iron absorption. Prolonged application of low-dose iron has been shown to reduce cortisol [41], but the correlating hormonal feedback regulation mechanisms have not been examined in detail. However, the present study concentrated on the effects of an iron injection as a single dosage often used in the field. The elevated cortisol levels in the animals of the present study result from acute stress due to birth, transportation, and disease. It is not predictable if higher single iron dosages or repeated application of iron might have altered the results.

Inflammatory processes in the calves were indicated by slightly increased WBC counts and TP concentrations in the diseased animals, which were not statistically greater than in healthy calves. The interpretation of these findings is complicated due to the physiological elevation of WBC counts during the first week of life [43] and the potential influence of colostrum quality on TP concentrations. The obtained results for WBC and TP in the present study align with physiological ranges previously published in dairy calves of one to nine days of age [44], verifying the mild character of WBC enhancement in the diseased calves.

Lactate concentrations increase in many diseases and are associated with inflammation [45,46]. Hyperlactatemia is associated with iron-deficiency anemia through the regulation of hepcidin expression in both mice and humans [46]; a marked impact of iron supplementation on lactate concentration was observed in the current trial. This apparent contradiction and the absence of elevated lactate concentrations in diseased animals in the current investigation could be clarified by the recent recognition of lactate as a dual regulator of cytokine expression and immune responses with both pro- and anti-inflammatory properties [45]. Due to the congruent timely development of lactate concentrations in supplemented and control calves, the elevated lactate concentrations in supplemented calves are regarded to be the result of iron-induced oxidative stress in those animals. To verify this assumption, classical markers of oxidative stress (i.e., ROS, malonaldehyde, or 8-hydroxy-2′-deoxyguanosine) should have been examined representing a limitation of the present study protocol.

Substance P (SP), a proinflammatory tachykinin, is well-documented in inflammatory diseases and pain evaluation [36,47,48,49,50]. As inflammation affects iron metabolism and serum ferritin concentrations [33,51,52], a potential correlation between SP concentrations and serum iron or ferritin concentrations in calves with inflammatory conditions was hypothesized. Consistent with the negative correlation observed between serum iron and serum amyloid A (SAA) concentrations in cows during dehorning [51], a weak negative correlation between ferritin and SP concentrations in the male calves was noted in the current study. Thus, higher SP concentrations were linked to slightly lower ferritin concentrations. Weak positive correlations between ferritin concentrations and SP in the control and female calves were also noted in the current study. These conflicting findings regarding SP may stem from significant interindividual variability and sex-dependent differences in this parameter [36,50].

Haptoglobin is a major acute-phase protein in cattle and has been shown to increase in response to infection, inflammation, or trauma [53]. However, when evaluating haptoglobin concentrations, it is important to consider the uncertainties surrounding the reference ranges. For instance, in adult cattle, haptoglobin concentrations ranging from >0.01 to >0.35 mg/mL have been linked to inflammatory conditions [54,55,56]. In dairy heifer calves, haptoglobin concentrations exceeding 0.13 mg/mL during the first week of life have been found to predict disease treatment and mortality up to four months of age [57]. The predictive value of haptoglobin for survival in systemic inflammatory response syndrome (SIRS) cases is not recommended due to the lack of a statistical difference in haptoglobin concentrations between calves with SIRS (Hp: 0.29 mg/mL; range, 0.05–3.6) and those without (0.22 mg/mL; range, 0–4.2) [58]. In the current study, haptoglobin concentrations in diseased calves did not exceed 0.35 ± 0.53 mg/mL by day 7 and began to differ from healthy calves starting from day 5, although not statistically. Whereas higher haptoglobin concentrations have been observed in young calves with diarrhea [59] or respiratory diseases [60], the anti-inflammatory properties of this acute-phase protein have also been documented [58]. These characteristics of haptoglobin align with our findings on the negative correlations between haptoglobin and serum iron, and haptoglobin and hepcidin. Higher concentrations of haptoglobin lead to decreased serum iron concentrations through the binding of hemoglobin to the haptoglobin molecule [58].

In addition to its role as the main intra- and extracellular storage protein for iron [6], ferritin functions as an acute-phase protein linked to inflammatory processes, and it plays a role in iron retention within the reticuloendothelial system [61]. A positive association with other proinflammatory acute-phase proteins, such as interleukin 1β (IL1β) and interleukin 6 (IL6), has been observed in viral and bacterial infections in humans, although the sensitivity and specificity of this relationship require further clarification [62]. Human serum ferritin concentrations have been suggested as inflammatory biomarkers associated with all-cause mortality with a cut-off value of 194 ng/mL [63]. However, corresponding cut-off values to identify fatal inflammatory states in cattle as well as reference ranges are lacking. The current study also identified positive correlations between ferritin and IL1β, as well as IL6. The control calves had higher IL1β and lower IL6 concentrations than the iron-supplemented calves. The correlation between ferritin and IL1β was consistent across groups and sexes, whereas the correlation between ferritin and IL6 was statistically significant only in the control group, unaffected by sex. These findings collectively suggest that elevated iron storage concentrations are associated with a pro-inflammatory state. The underlying mechanism of elevated iron storage is a hepcidin-mediated internalization of iron into enterocytes and macrophages. Subsequently, expression of hepcidin is mediated by cytokines like IL1β and IL6 [6]. Further, iron overload may lead to the formation of ROS, thus, causing damage to tissues and enzymes and activating inflammatory pathways [6].

The results indicate a diminished antioxidant capacity in the calves in this study. Elevated ferritin concentrations indicate increased iron storage and produce a negative association between inflammation and FRAP values. Animals with reduced antioxidant capacity may have an increased susceptibility to infection. The FRAP values in the current study were lower than those documented by Albera et al. [64]. A direct comparison of FRAP concentrations between our study and [64] is not feasible. Nevertheless, there is some agreement with the work of Albera et al. [64], which demonstrated a decline in antioxidant capacity postpartum. This concurs with the observations during the first 10 d of the current study, further supported by a negative correlation with serum ferritin concentrations. Our findings of a reduced antioxidant capacity are also in line with Fu et al. [65], who could show a reduced total antioxidant capacity in diarrheic calves. However, it is not clear whether the altered antioxidative status is the cause or the result of diarrhea. Nonetheless, the results of the present studies suggest and explain the contraindication of iron supplementation in inflammatory diseases (i.e., diarrhea). Cytokine concentrations and enzyme activities are altered in ailing calves [65], thus, representing a reduced capacity to disarm iron overload which in turn may cause tissue and enzyme damage via the production of ROS. Therefore, general iron supplementation treatments in calves with inflammatory diseases should be rejected.

Diagnostic parameters for iron-deficiency anemia include serum iron, serum ferritin, transferrin saturation, hepcidin concentration, and the expression of its receptor, ferroportin [66,67]. Hepcidin is mainly produced by hepatocytes and is crucial for maintaining stable plasma iron concentrations and storage in the body’s iron reserves [66,67]. Its concentration varies substantially within populations due to circadian variations [67,68]. Determining hepcidin concentration is challenging due to considerable variation and pre-analytical issues, such as tube adhesion and protein aggregation in media. Techniques such as surface-enhanced laser desorption/ionization time-of-flight mass spectrometry (TOF-MS), matrix-assisted laser desorption/ionization TOF-MS, liquid chromatography-tandem mass spectrometry [67], and enzyme-linked immunosorbent assays have been used to measure hepcidin concentrations and establish age- and sex-specific reference ranges [68,69]. However, conflicting reports exist regarding the relationship between serum ferritin and hepcidin concentrations. Although Galesloot et al. [69] and Wolff et al. [67] report strong correlations, Taheri et al. [68] found no association between serum ferritin and hepcidin concentrations. In the present study, no correlation between the AUC of serum ferritin and hepcidin was observed, but an inconsistent correlation between the AUC of serum iron and hepcidin, influenced by sex and day, was determined, concurring with the aforementioned concentration variances.

A general limitation of our study is that the correlations were rather weak. This could be attributed to perinatal adjustment processes and age-dependent concentration changes related to enzymes and hepatic hemodynamic changes until early adulthood [43,52,70]. In comparison to other studies [1,2,3,7], iron dosage, route of administration, and time of treatment may have also influenced the results. Our study used a rather low dosage of iron dextran of 10 mg/kg to mimic the dosage often used in standard procedures in the field. Application of higher iron dosages or repeated treatments might have resulted in stronger correlations between the examined parameters. Meanwhile, the biological relevance of existing correlations and their underlying effect sizes have to be taken into account. Despite being weak, the consistent correlations between ferritin and markers of inflammation presented in our study are suggested to encourage veterinarians in the field to critically overthink iron supplementations as a standard treatment procedure in neonatal calves. Another crucial aspect and limitation of the present study may have been the 10 d observation period. A more extended examination might have yielded different patterns in the concentrations of the parameters under study. Furthermore, the calves were sourced from various local farms, leading to heterogeneity in fetal iron supply and immune status. The quality of colostrum was not examined on the farms, which may have introduced variability in the passive transfer of immunoglobulins, potentially comprising immune defense in the calves included in our study. The development of various diseases and the heterogenous fetal iron supply may have impacted the study outcomes.

Due to continuous herd health control practices on the farms of origin, the calves with diarrhea underwent parasitological examinations only. Other potential infectious agents, such as *Escherichia coli*, *Salmonella* spp., rotavirus, or coronavirus, were considered circumstantial. Conducting specific microbiological tests would have been justified to evaluate the possible effects of iron-using bacteria along with the impact of compromised intestinal iron absorption resulting from enteritis.

Despite these limitations, the present results offer deeper insights into the relationship between iron metabolism and inflammation in neonatal calves.

## 5. Conclusions

Cortisol concentrations were higher immediately postpartum, declining during the first 10 d of life. The impact of iron therapy on cortisol concentrations remained indeterminate despite higher cortisol concentrations in ailing calves than in healthy ones. Serum ferritin exhibited weak positive associations with the proinflammatory cytokines, IL1β and IL6 while displaying negative correlations with FRAP, indicative of diminished antioxidative capacity. Diseased calves consistently demonstrated elevated WBC counts and TP concentrations throughout the observation period, with no marked variations based on treatment, illness, or sex of the animals. Lactate concentrations were elevated in calves that were administered iron supplements.

This study is the first to demonstrate the impact of inflammatory diseases on serum iron and cortisol concentrations in neonatal calves. In the future, the clinical significance and resulting therapeutic consequences of serum ferritin, transferrin saturation, and hepcidin concentrations should be further explored in calves, young cattle, and adult cattle. Future attempts to establish reliable reference ranges are highly encouraged to enable therapeutic consequences. Nevertheless, no unthinking iron supplementation should be performed in animals suffering from inflammatory diseases like diarrhea, omphalitis, or pneumonia.

## Figures and Tables

**Figure 1 animals-15-01021-f001:**
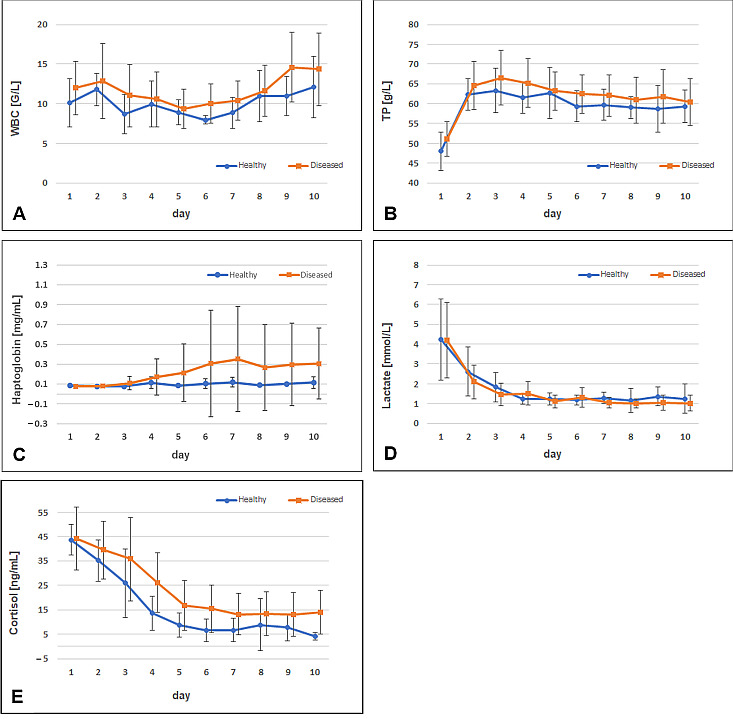
Comparison of the time patterns of white blood cell (WBC) counts (**A**), total protein (TP) (**B**), haptoglobin (**C**), lactate (**D**), and cortisol concentrations (**E**) between healthy and diseased calves. Dots and squares indicate arithmetic means, with whiskers representing standard deviations.

**Figure 2 animals-15-01021-f002:**
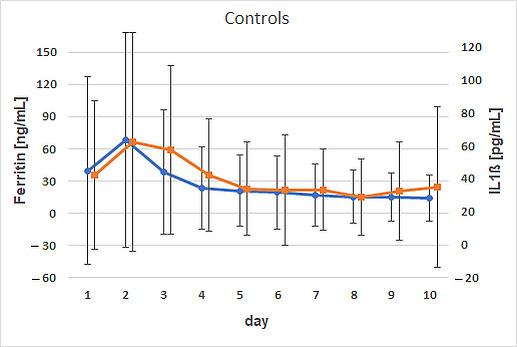
Positive correlations between the time courses of ferritin and IL1β on days 2 and 3 in calves that received no iron supplementation (controls). Dots (ferritin) and squares (IL1β) represent arithmetic means, with whiskers representing standard deviations.

**Figure 3 animals-15-01021-f003:**
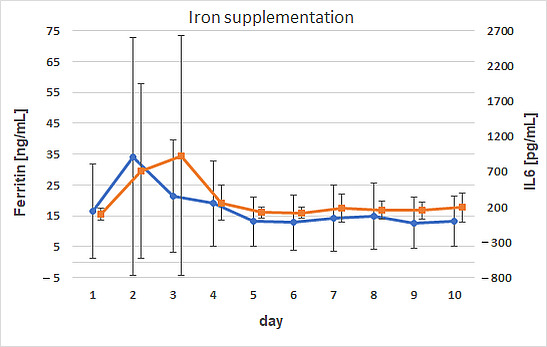
Comparison of the time courses of ferritin and IL6 for calves supplemented with iron. Dots (ferritin) and squares (IL6) represent arithmetic means; whiskers represent standard deviations.

**Figure 4 animals-15-01021-f004:**
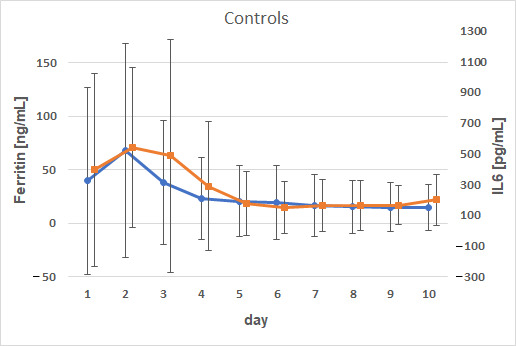
Comparison of the time courses of ferritin and IL6 for calves without iron supplementation. Dots (ferritin) and squares (IL6) represent arithmetic means; whiskers represent standard deviations.

**Figure 5 animals-15-01021-f005:**
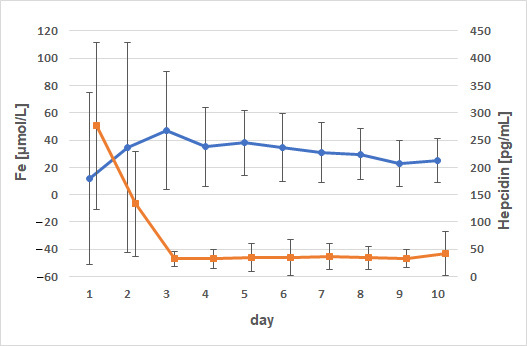
Comparison of the time courses of serum iron and hepcidin. Dots (Fe) and squares (hepcidin) represent arithmetic means; whiskers represent standard deviations; negative correlations were observed.

**Figure 6 animals-15-01021-f006:**
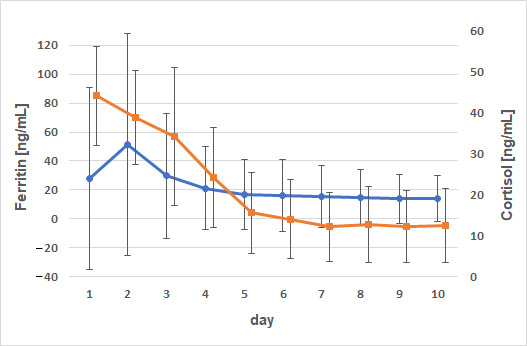
Comparison of time courses of ferritin and cortisol concentrations ($\overset{⇁}{x}$ ± s; negative correlation particularly in male animals with *p* = 0.0009 and ρ = −0.68). Dots (ferritin) and squares (cortisol) represent arithmetic means; whiskers represent standard deviations.

**Figure 7 animals-15-01021-f007:**
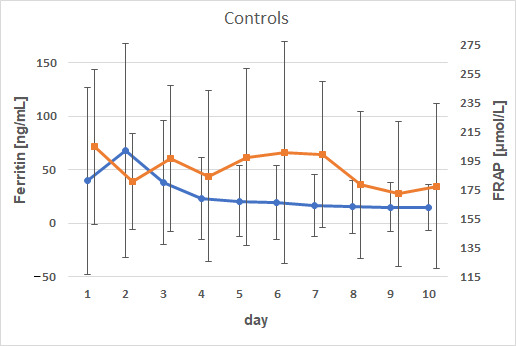
Weak negative correlations between ferritin (dots) and FRAP (squares) in calves of the control group ($\overset{⇁}{x}$ ± s; *p* = 0.0223; ρ = −0.50771).

**Table 1 animals-15-01021-t001:** *p*-values and Spearman correlation coefficients for haptoglobin correlation analyses. The results of the correlation analyses for haptoglobin depended on treatment and/or sex.

Variable	Grouping by Sex or Treatment	*p*-Value	Spearman Correlation Coefficient
Fe	Treatment: Yes	0.0024	−0.64
Fe	Sex: Female	0.0021	−0.65
Hepcidin	Treatment: No	<0.0001	−0.83
Hepcidin	Sex: Female	0.0016	−0.66
Hepcidin	Sex: Male	0.0046	−0.61
Lactate	Treatment: Yes	0.0386	−0.47
IL1β	Sex: Male	0.0427	−0.46

## Data Availability

The datasets used and/or analyzed in the current study are available from the corresponding author upon reasonable request.

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
