# Peer review of "Association of Stress and Inflammatory Diseases with Serum Ferritin and Iron Concentrations in Neonatal Calves"

_animals, 2025, doi:10.3390/ani15071021_

Round 1

Reviewer 1 Report

Comments and Suggestions for Authors

Line 18 – Is Fe supplementation in calves, possibly dairy calves, really performed on the first day of life? It is usually done in the first week, as stated in the text below.

Line 50 – Which hormone or hormones are involved in Fe metabolism?

Lines 55–57 - What are the main enzymes involved?

Lines 73–83 – Discuss the biological function of ferritin.

Lines 80–83 – Explain how pro-inflammatory cytokines influence the plasma concentration of Fe and ferritin.

Section 2.1 – I believe that the period of only 10 days is too short, because despite the conditions of minimal viral and bacteriological risk, the animals still do not reach the risk zone for pneumonia, that is, around 40 days of life. Therefore, the evaluation period should be extended or the experiment should be carried out with a later second phase.

Were the animals' navels treated with iodine, for example, or not? If not, was this intentional? Considering the high prevalence of omphalophlebitis.

Lines 100-101 – Provide better details on the colostrum dosage. Since, from 2 to 3 liters, there is a 50% increase.

There was a prior evaluation of the colostrum offered, since this will greatly influence the individual's immune response capacity at this time.

Line 156 – What congenital anomalies?

Section 4 – Pay attention to the text formatting.

Briefly discuss the biological functions of the following parameters evaluated: IL1β, IL6, SP, hepcidin, haptoglobin and FRAP. Since this can provide greater clarity to the results obtained.

Author Response

Dear Editor and Reviewers,

The authors would like to thank you for the evaluation of our manuscript, and your time and efforts to provide constructive comments and remnant concerns. You have pointed out important aspects, which have been taken into careful consideration. The authors hope that the edits made are to your satisfaction and that the amended manuscript will be considered for publication in Animals.

“Your manuscript has now been reviewed by experts in the field and can be found with the review reports at: https://susy.mdpi.com/user/manuscripts/resubmit/716cb0e5e61ad56c1b951fbccff627b7

Please revise the manuscript found at the above link according to the reviewers' comments and upload the revised file within 10 days.”

The authors have addressed each of the comments below.

Reviewer 1

Open Review

Quality of English Language

( ) The English could be improved to more clearly express the research.
(x) The English is fine and does not require any improvement.

Yes

Can be improved

Must be improved

Not applicable

Does the introduction provide sufficient background and include all relevant references?

(x)

( )

( )

( )

Is the research design appropriate?

( )

( )

(x)

( )

Are the methods adequately described?

( )

(x)

( )

( )

Are the results clearly presented?

(x)

( )

( )

( )

Are the conclusions supported by the results?

(x)

( )

( )

( )

Answer:

Thank you very much for your time and effort in reading our manuscript and giving this constructive feedback. We appreciate your criticism and are sure that your advice will improve the quality and readability of our paper.

Comments and Suggestions for Authors

Line 18 – Is Fe supplementation in calves, possibly dairy calves, really performed on the first day of life? It is usually done in the first week, as stated in the text below.

Answer:

Our experience in this point is that iron supplementation is performed right after birth as a first-day booster treatment along with feeding of colostrum directly after birth. The calves that were included in our study originated from farms localized within a few kilometers from our clinic. The standard iron supplementation regime in our catchment area is to treat the calves directly after birth. However, a more suitable time point for treatment will be discussed with our clients after performing this study. To harmonize the wording of the paper, we have changed the lines 17–19 as follows: “Iron supplementation is commonly administered within the first days of life to prevent anemia.”

Line 50 – Which hormone or hormones are involved in Fe metabolism?

Answer:

In lines 50, we refer to a “hormone-like” mechanism that regulates iron homeostasis. The relevant peptides that are involved in this mechanism are hepcidin and ferroportin. Ferroportin represents the basolateral iron transport channel in enterocytes for transportation of Fe2+ from the intracellular to extracellular region. This channel contains a receptor for hepcidin. After binding of hepcidin to its receptor, the ferroportin channel is internalized into the enterocyte resulting in a reduction of iron transfer to the plasma. The regulating peptide hepcidin is synthetized within the liver as soon as iron-binding to transferrin has taken place within the blood. High iron concentrations in the blood lead to binding of iron-transferrin complexes to special receptors within the hepatocytes which in turn results in an enhanced production of hepcidin within the liver. This feedback-mechanism for the iron resorption system and homeostasis resembles the feedback mechanisms of many hormones and is therefore, regarded as a hormone-like mechanism (cited according to: Humann-Ziehank E. Neuer Blick auf ein altes Element – Eisen, hepcidin und Entzündung [New insights on a long-known element – iron, hepcidin and inflammation]. Tierarztl Prax Ausg G Grosstiere Nutztiere (2020) 48:183–90. doi: 10.1055/a-1162-0126).

However, there are several classical hormones (e.g. adrenaline, noradrenalin, and estrogens) that are also capable of influencing the iron resorption and excretion mechanisms.

We thought that such explanations would unnecessarily elongate the introduction section, but if the reviewer would like us to include a passage referring to the iron regulating mechanisms within the introduction we will be happy to do so.

Lines 55–57 - What are the main enzymes involved?

Answer:

We are deeply regretful yet uncertain concerning the premise of this inquiry. Lines 55–77 refer to the potentially harmful consequences of an iron overload. This refers to the high potential of iron to act as electron donors and acceptors, and to bind to several ligands such as oxygen, nitrogen or sulfur. Especially the capability to change easily between the oxidative forms of Fe2+ and Fe3+ is responsible for the formation of reactive oxygen species that may affect biological membranes and even cells.

However, in trying to understand the target course of the reviewer’s question, we have spotted an error in the original manuscript draft and changed the following: hemoglobin and myoglobin are non-enzymatic and iron is part of physiological enzymes like cytochroms or iron-sulfur-enzymes (e. g. Succinatdehydrogenase). We have therefore changed the wording of lines 53–54.

Lines 73–83 – Discuss the biological function of ferritin.

Answer:

The biological function of ferritin is complex, but its most relevant biological role in the context of our manuscript is the intracellular and extracellular storing of iron. Due to the high potential of iron to react as a redox partner and form ROS, intracellular storage and disarming of iron is essential (Humann-Ziehank). Hepatocytes, macrophages, and Kupffer cells are able to secrete ferritin, which enables them to both store iron and make it available for cells. The mechanism for making iron accessible for cells relies on the presence of a saturable ferritin receptor on the surface of different cell types like lymphocytes, hepatocytes, and oligodendrocytes (Wang et al.). Serum ferritin concentrations are correlated to the intracellular iron contents, thus, representing a measure for the iron storage of the organism (Daru et al.). However, serum ferritin is also regarded as an acute phase protein that is nonspecifically enhanced during inflammatory conditions and representing elevated iron storages in such cases. However, during inflammation, stored iron is sequestered and is unavailable for hematopoiesis being the underlying correlation for the so-called anemia of inflammation (Wang et al.)

To give a short insight into the biological function of ferritin within the introduction, we have included the following (lines 77–81): “Ferritin serves as the main intra- and extracellular storage protein for iron [6]. Several cells like hepatocytes, macrophages, and Kupffer cells are capable of secreting ferritin, which is not only able to store iron but also make it available for cells [23]. Serum ferritin concentrations are correlated to the intracellular iron contents, thus, representing a measure for the iron storage of the organism [21].”

Lines 80–83 – Explain how pro-inflammatory cytokines influence the plasma concentration of Fe and ferritin.

Answer:

Pro-inflammatory cytokines like IL-1β, IL6, and TNFα influence plasma concentration of Fe and ferritin by inducing the secretion of ferritin from hepatocytes. The relative ratio of ferritin to iron is modulated by an enhanced hepcidin expression which consequently reduces the iron absorption leading to reduced serum iron concentrations in the status of inflammation (Ueda et al., 2018). To explain the aforementioned mechanisms, we have added the following to the manuscript (lines 89–96): “Proinflammatory cytokines like IL-1β, IL6, and TNFα lead to an enhanced secretion of ferritin from hepatocytes and an enhanced expression of hepcidin. Thus, iron absorption is reduced and the relative ratio of ferritin to iron is modulated in animals affected by inflammatory diseases [28]. However, serum ferritin itself is also regarded as an acute phase protein that is nonspecifically enhanced during inflammatory conditions and represents elevated iron storages in such cases. In such cases, the stored iron is sequestered and is not available for hematopoiesis being the underlying correlate for the so-called anemia of inflammation [23]” (Wang et al., 2010)

Section 2.1 – I believe that the period of only 10 days is too short, because despite the conditions of minimal viral and bacteriological risk, the animals still do not reach the risk zone for pneumonia, that is, around 40 days of life. Therefore, the evaluation period should be extended or the experiment should be carried out with a later second phase.

Answer:

Thank you very much for your insightful comment. However, the aim of our study was to examine the developments and mutual relationships of blood parameters with iron supplementation and diseases of neonatal calves. The neonatal stage is generally regarded to last for the first 14 days of life. Several previous studies in calves (1), piglets (2), and humans (3) could show a supplementation-associated enhancement of serum ferritin within the first week of life. Therefore, we were confident, that the examination period of 10 days in our calves would allow us to gain valid data. Other studies like Golbeck et al. (PeerJ. 2019;7:e7248. doi:10.7717/peerj.7248) concentrated on older calves, but no data concerning the effects of supplementation and inflammation was present, especially in very young calves. Nevertheless, we agree with the reviewer that a second phase of evaluation would have been interesting. Therefore, we had included this limitation in lines 568–571 of the manuscript. Please kindly refer to this section.

  • Miyata Y, Furugouri K, Shijimaya K. Developmental changes in serum ferritin concentration of dairy calves. J Dairy Sci. 1984;67:1256-63. doi:10.3168/jds.S0022-0302(84)81432-0.
  • Furugouri K, Miyata Y, Shijimaya K, Narasaki N. Developmental changes in serum ferritin of piglets. J Anim Sci. 1983;57:960-5.
  • Wheby MS. Effect of iron therapy on serum ferritin levels in iron-deficiency anemia. Blood 1980;56:138-140.

Were the animals' navels treated with iodine, for example, or not? If not, was this intentional? Considering the high prevalence of omphalophlebitis.

Answer:

As a standard procedure, the animals’ navels were treated with iodine and we have included this information in lines 144/145.

Lines 100-101 – Provide better details on the colostrum dosage. Since, from 2 to 3 liters, there is a 50% increase.

Answer:

Thank you for your query. All calves received two colostrum meals with colostrum from

their dams. The amount of colostrum fed was based on the body weight of the calves, with

at least 6 L of colostrum fed during the first 12 h of life. Accordingly, we have adapted the manuscript in lines 120–121.

There was a prior evaluation of the colostrum offered, since this will greatly influence the individual's immune response capacity at this time.

Answer:

We are completely in agreement with the reviewer’s evaluation of a possible influence of colostrum quality on the immune response of the calves. Therefore, we had already included this limitation in the discussion section. Please refer to lines 572–575.

Line 156 – What congenital anomalies?

Answer:

At our clinic, the standard examination protocol in neonates includes explicit examination for the presence of obvious congenital anomalies like palatine clefts or atresia ani. To clarify the same, we have adapted the manuscript at lines 197–198.

Section 4 – Pay attention to the text formatting.

Answer:

We thank the reviewer for also paying attention to the general aspects of the manuscript, and we have gladly adapted and harmonized the formatting of the discussion section. We expect the revised manuscript to comply with all text formatting requirements.

Briefly discuss the biological functions of the following parameters evaluated: IL1β, IL6, SP, hepcidin, haptoglobin and FRAP. Since this can provide greater clarity to the results obtained.

Answer:

We are very grateful to the reviewer for this valuable suggestion. We have included more information concerning the proinflammatory cytokines IL-1β and IL6 in the introduction (lines 89–96). Concerning SP, hepcidin, haptoglobin, and FRAP, their biological functions are given in the discussion section within the relevant paragraphs. We hope that the information included finds the reviewer’s approval and satisfaction.

Submission Date

06 March 2025

Date of this review

13 Mar 2025 15:28:28

We would like to thank the reviewer again for the effort and time taken in reviewing our manuscript. We highly appreciate the acknowledgement of our attempts to fulfill your recommendations and to address your criticism. Your comments greatly contributed to the improvement of our manuscript.

Reviewer 2 Report

Comments and Suggestions for Authors

This study investigated the association between iron metabolism and inflammatory/stress indicators in neonatal calves, analyzing the effects of iron supplementation, disease status, and sex on relevant parameters. The research topic holds clinical significance, and the experimental design is generally reasonable with comprehensive data. However, methodological details, result interpretation, and discussion depth require further refinement to enhance scientific rigor. Specific comments are as follows: 

nsufficient Control of Sample Heterogeneity

  1. Colostrum quality: Calves were sourced from different farms, but colostrum quality (e.g., IgG concentration) was not measured, potentially affecting the homogeneity of immune status and iron metabolism. Supplemental colostrum quality data or discussion of its potential impact on results is needed.
  2. Environmental and management factors: Descriptions of environmental conditions (temperature, humidity, ventilation) and feeding management details (e.g., milk feeding frequency) are lacking, which may introduce confounding factors.

Unclear Iron Supplementation Protocol

  1. Dosage rationale: The basis for selecting the iron dextran dose (10 mg/kg) was not explained (e.g., pharmacokinetic studies or industry standards).
  2. Metabolic impact: The direct effects of iron supplementation on other metabolic parameters (e.g., oxidative stress markers) were not evaluated, making it difficult to distinguish between the independent effects of iron supplementation and inflammation.

Insufficient Validation of Detection Methods

  1. ELISA reliability: Intra- and inter-assay coefficients of variation for ELISA detection of inflammatory factors (e.g., IL1β, IL6) were not provided; supplemental data are required to confirm reliability.
  2. Specificity of cortisol RIA: Validation of the cortisol radioimmunoassay (RIA) in bovine samples was insufficient (e.g., cross-reactivity with common interferents).

Inadequate Mechanistic Exploration

  1. Ferritin-cytokine correlations: The positive correlations between ferritin and IL1β/IL6 were solely attributed to a "pro-inflammatory state" without clarifying causality (e.g., whether inflammation drives increased iron storage or iron overload directly activates inflammatory pathways).
  2. Cortisol-iron metabolism link: The association between cortisol and iron metabolism lacks mechanistic explanation (e.g., whether the HPA axis regulates iron metabolism via hepcidin).

Discussion 

  1. Lactate elevation mechanism: The potential mechanism for increased lactate in iron-supplemented calves (e.g., iron-induced oxidative stress or energy metabolism alterations) was not explored.
  2. Sex-specific SP correlations: The opposite correlations between substance P (SP) and ferritin in male vs. female calves require hypotheses integrating sex-specific hormonal or immune differences.

Insufficient Literature Comparison

  1. Lack of calf study comparisons: Comparisons with similar calf studies (e.g., whether the physiological elevation window of WBC aligns with literature [36]) are missing, weakening result comparability.
  2. Human-ferritin thresholds: The applicability of human ferritin thresholds for inflammation (e.g., >194 ng/mL) to calves was not discussed.

Weak Clinical Translation

  1. Diagnostic/practical gaps: No clinical diagnostic threshold recommendations based on ferritin or FRAP were proposed, nor were optimized iron supplementation strategies (e.g., dose adjustments by disease status) suggested.

Author Response

Dear Editor and Reviewers,

The authors would like to thank you for the evaluation of our manuscript, and your time and efforts to provide constructive comments and remnant concerns. You have pointed out important aspects, which have been taken into careful consideration. The authors hope that the edits made are to your satisfaction and that the amended manuscript will be considered for publication in Animals.

“Your manuscript has now been reviewed by experts in the field and can be found with the review reports at: https://susy.mdpi.com/user/manuscripts/resubmit/716cb0e5e61ad56c1b951fbccff627b7

Please revise the manuscript found at the above link according to the reviewers' comments and upload the revised file within 10 days.”

The authors have addressed each of the comments below.

Reviewer 2

Open Review

Quality of English Language

( ) The English could be improved to more clearly express the research.
(x) The English is fine and does not require any improvement.

Yes

Can be improved

Must be improved

Not applicable

Does the introduction provide sufficient background and include all relevant references?

(x)

( )

( )

( )

Is the research design appropriate?

(x)

( )

( )

( )

Are the methods adequately described?

( )

(x)

( )

( )

Are the results clearly presented?

( )

(x)

( )

( )

Are the conclusions supported by the results?

( )

(x)

( )

( )

Comments and Suggestions for Authors

This study investigated the association between iron metabolism and inflammatory/stress indicators in neonatal calves, analyzing the effects of iron supplementation, disease status, and sex on relevant parameters. The research topic holds clinical significance, and the experimental design is generally reasonable with comprehensive data. However, methodological details, result interpretation, and discussion depth require further refinement to enhance scientific rigor. Specific comments are as follows:

Answer:

Thank you very much for your time and effort in reading our manuscript and giving this constructive feedback. We appreciate your criticism and are confident that your advice will improve the quality and readability of our paper. In the following, we will address the specific comments made.

Insufficient Control of Sample Heterogeneity

  1. Colostrum quality: Calves were sourced from different farms, but colostrum quality (e.g., IgG concentration) was not measured, potentially affecting the homogeneity of immune status and iron metabolism. Supplemental colostrum quality data or discussion of its potential impact on results is needed.

Answer:

We are completely in line with the reviewer’s opinion that colostrum quality has a direct impact on the immune status of the animals, and can thus influence the results. Therefore, we have referred to this point as a limitation of the study. Kindly refer to lines 572–575 of the manuscript for more details.

  1. Environmental and management factors: Descriptions of environmental conditions (temperature, humidity, ventilation) and feeding management details (e.g., milk feeding frequency) are lacking, which may introduce confounding factors.

Answer:

The animals were housed within the facilities of the Clinic for Ruminants. Climate conditions depended on the surrounding climate. We have included this information in the manuscript (lines 124–126). For details concerning feeding management, we kindly ask the reviewer to refer to lines 127–130: “A second colostrum meal was provided within the first 6 h of birth, followed by feeding with a commercial milk substitute three times daily (CombiMilk Start, Agravis Raiffeisen AG; Fe(II): 150 mg/kg DM). Hay, concentrate feed (Kälberkraft; Agravis Raif-feisen AG), and water were freely available from the second day of birth.”

Unclear Iron Supplementation Protocol

  1. Dosage rationale: The basis for selecting the iron dextran dose (10 mg/kg) was not explained (e.g., pharmacokinetic studies or industry standards).

Answer:

To clarify the rationale for selection of the iron dextran dosage, we have added the following sentence: “This dosage represents the standard dosage recommended by the manufacturer and is in line with the standard protocols used in the field within the catchment area of the clinic.” (lines 149151)

  1. Metabolic impact: The direct effects of iron supplementation on other metabolic parameters (e.g., oxidative stress markers) were not evaluated, making it difficult to distinguish between the independent effects of iron supplementation and inflammation.

Answer:

We are in complete agreement with the reviewer on this point. However, our study was designed to investigate the timely development of several blood parameters including alterations of RBC and erythrocyte indices (Sickinger et al., 2024) as well as the influence of supplementation and disease on stress parameters like cortisol and lactate, and on inflammation markers like cytokines and SP. Nevertheless, you are correct in noting that oxidative stress markers should have been included. This is clearly a limitation of the study and we have included this point into the discussion section (lines 461–466 referring to lactate), because it overlaps with the topic of question 9. We hope that we could meet the intended request of the reviewer on this point.

Insufficient Validation of Detection Methods

  1. ELISA reliability: Intra- and inter-assay coefficients of variation for ELISA detection of inflammatory factors (e.g., IL1β, IL6) were not provided; supplemental data are required to confirm reliability.

Answer:

We regret that this information was missing within the initially submitted manuscript version. We have included the relevant information in the materials and methods section (lines 177–178, 180–181).

  1. Specificity of cortisol RIA: Validation of the cortisol radioimmunoassay (RIA) in bovine samples was insufficient (e.g., cross-reactivity with common interferents).

Answer:

We are regretful to state that we are not sure what the reviewer means by “common interferents.” The manuscript contains cross-reactivity data concerning cortisol, corticosterone, androstenedione, 5α-dihydrotestosterone, estradi-ol-17β, pregnenolone, progesterone, and testosterone. Please kindly refer to lines 188–192 for details. If there should be any further data missing, we would be glad to address concrete questions on this topic.

Inadequate Mechanistic Exploration

  1. Ferritin-cytokine correlations: The positive correlations between ferritin and IL1β/IL6 were solely attributed to a "pro-inflammatory state" without clarifying causality (e.g., whether inflammation drives increased iron storage or iron overload directly activates inflammatory pathways).

Answer:

We thank the reviewer for this insightful comment. We have attempted to clarify the causality of our findings by introducing the following wording to the discussion section: “The underlying mechanism of elevated iron storage is a hepcidin-mediated internalization of iron into enterocytes and macrophages. Subsequently, expression of hepcidin is mediated by cytokines like IL1β and IL6 [6]. Further, iron overload may lead to the formation of ROS, thus, causing damage to tissues and enzymes and activating inflammatory pathways [6].” (lines 514–518)

  1. Cortisol-iron metabolism link: The association between cortisol and iron metabolism lacks mechanistic explanation (e.g., whether the HPA axis regulates iron metabolism via hepcidin).

Answer:

There is no clear evidence that the HPA axis is involved in the linkage between iron metabolism and cortisol concentrations. It is suspected that this relation is based on indirect regulation via the promotion of inflammation in acute stress (higher cortisol concentrations in the diseased calves) being associated with an enhanced production of hepcidin within hepatocytes. To clarify this uncertainty, we have adapted the manuscript draft and included the following passage: “Significance might have been reached using higher dosages of iron or a prolonged oral supplementation of iron and selenium as has been shown in human athletes [40] and young bulls transported over long-distances [41]. In athletes, a reduction of cortisol was achieved after oral supplementation of iron over a period of 3 weeks. In bulls, the application of slow-release selenium boluses resulted in an elevation of serum iron and a reduction of cortisol. The link between cortisol and iron metabolism is regarded to be an indirect effect of a promotion of inflammation in acute stress (i.e. higher cortisol concentrations in diseased animals). In animals suffering from acute stress, proinflammatory cytokines are enhanced leading to higher concentrations of hepcidin [6], thereby reducing iron absorption. Prolonged application of low-dose iron has been shown to reduce cortisol [40], but the correlating hormonal feedback regulation mechanisms have not been examined in detail. However, the present study concentrated on the effects of an iron injection as a single dosage often used in the field. The elevated cortisol levels in the animals of the present study result from acute stress due to birth, transportation, and disease. It is not predictable if higher single iron dosages or repeated application of iron might have altered the results.” (lines 430–445)

Discussion

  1. Lactate elevation mechanism: The potential mechanism for increased lactate in iron-supplemented calves (e.g., iron-induced oxidative stress or energy metabolism alterations) was not explored.

Answer:

We concur with the reviewer that our study did not include a specific examination concerning classical oxidative stress markers like ROS, malonaldehyde (MDA), and 8-hydroxy-2′-deoxyguanosine (8-OHDG). Clearly, this is a limitation to our study, but was due to a very intense sampling frequency in the animals, which restricted the amount of blood that could be taken. Therefore, we had to favor some parameters over others and evaluated the overall anti-oxidant capacity by determining FRAP. Due to the congruent development of lactate concentrations in supplemented and control calves, we estimated the elevated lactate concentrations in iron supplemented calves as the result of iron-induced oxidative stress rather than being the correlate of changes in energy metabolism.

We have included the following paragraph in the discussion section to critically emphasize the limitation of our study concerning an evaluation of the extent of oxidative stress:

“Due to the congruent timely development of lactate concentrations in supplemented and control calves, the elevated lactate concentrations in supplemented calves is regarded to be the result of an iron-induced oxidative stress in those animals. To verify this assumption, classical markers of oxidative stress (i.e. ROS, malonaldehyde, or 8-hydroxy-2′-deoxyguanosine) should have been examined representing a limitation of the present study protocol.” (lines 461–466)

  1. Sex-specific SP correlations: The opposite correlations between substance P (SP) and ferritin in male vs. female calves require hypotheses integrating sex-specific hormonal or immune differences.

Answer:

The presence of sex-specific differences in serum SP concentrations have been shown in calves (Landinger et. al, 2024). Due to the biological effects of SP as a neuromediator of pain, inflammation, and gastrointestinal motility, the possible sex-specific influences on SP concentrations may include more than simple hormonal differences. However, data verifying this suggestion are lacking for cattle. In humans and mice, the mechanisms of sex differences in immune response including the expression of several cytokines have only recently been described (Dunn et al., 2024). The authors state: “The broad distribution of sex steroid hormone receptors across diverse cell types and the differential expression of X-linked and autosomal genes means that sex is a biological variable that can affect the function of all physiological systems, including the immune system.” This is considered to be true for all mammals, thereby implicating that sex-specific differences should be considered in all examinations concerning biological systems.

To emphasize this fact, we have included a passage dealing with sex-specific effects on parameters in general (lines 406–420) and we hope that this information given in general meets the reviewer criticism.

Insufficient Literature Comparison

  1. Lack of calf study comparisons: Comparisons with similar calf studies (e.g., whether the physiological elevation window of WBC aligns with literature [36]) are missing, weakening result comparability.

Answer:

Thank you very much for indicating the missing literature comparison. We have included the study of Panousis et al. (2018) dealing with a comparable collective of animals (i.e. dairy calves aged 1 to 9 days) and compared our findings to the reference ranges recommended in this work (lines 450–453).

  1. Human-ferritin thresholds: The applicability of human ferritin thresholds for inflammation (e.g., >194 ng/mL) to calves was not discussed.

Answer:

The cut-off value of >194 ng/mL refers to a fatal outcome of several diseases in humans. To the best of our knowledge, no such cut-off value has been determined for cattle. There is also a lack for reference ranges for cattle. The establishment of reference ranges is recommended for the future. Concerning the present lack of knowledge, we have added the following wording to the manuscript: “However, corresponding cut-off values to identify fatal inflammatory states in cattle as well as reference ranges are lacking.” (lines 507­–508)

Weak Clinical Translation

  1. Diagnostic/practical gaps: No clinical diagnostic threshold recommendations based on ferritin or FRAP were proposed, nor were optimized iron supplementation strategies (e.g., dose adjustments by disease status) suggested.

Answer:

We understand the reviewer’s claim to gain recommendations concerning iron supplementation strategies and the establishment of reference ranges for ferritin and FRAP for example. To establish such reference ranges, large-scale studies would be needed and are highly recommended. The present study aimed rather to gain further insight into possible associations of ferritin and iron supplementation to inflammation and disease. We concur with the reviewer; future studies should concentrate on the establishment of reference ranges depending on age, sex, and disease. To emphasize this lack of knowledge, we have added the following wording to the conclusion section: “Future attempts to establish reliable reference ranges are highly encouraged to enable therapeutic consequences. Nevertheless, no unthinking iron supplementation should be performed in animals suffering from inflammatory diseases like diarrhea, omphalitis, or pneumonia.” (lines 599–602)

Submission Date

06 March 2025

Date of this review

15 Mar 2025 03:54:36

We would like to thank the reviewer again for the substantial suggestions and the critical remarks that definitely helped to improve the quality and depth of our present manuscript. It was a pleasure working with you on this draft and we are looking forward to further questions and remarks.

Reviewer 3 Report

Comments and Suggestions for Authors

The study investigates the interplay between iron metabolism, stress, and inflammatory markers in neonatal calves, addressing a relevant topic in veterinary physiology. While the research design is methodical and the data provide valuable insights, several areas require improvement to enhance clarity, statistical robustness, and clinical relevance.

Grammar and Syntax
The manuscript is generally well-written, but some sentences are overly complex or lengthy, reducing readability. For example:

  • "Neither iron supplementation, disease status, nor sex had statistically significant effects on the areas under the curve of ferritin, WBC, TP, IL1β, IL6, SP, hepcidin, haptoglobin, or FRAP."
    This sentence could be split for clarity. Minor grammatical errors (e.g., inconsistent tense usage in the Results section) and punctuation issues (e.g., missing commas in lists) are present but do not severely impede comprehension.

Structure and Content

  1. Introduction: Adequately contextualizes the study, but the mechanistic link between stress, inflammation, and iron metabolism could be expanded. The rationale for focusing on cortisol and specific cytokines (e.g., IL1β, IL6) is clear, but a brief explanation of why these markers were prioritized over others would strengthen the section.

  2. Materials and Methods:

    • The sample size (n=40) may limit statistical power, particularly for subgroup analyses (e.g., sex-specific effects). Justification for the sample size or a post-hoc power analysis is lacking.

    • Detailed descriptions of quality control measures for assays (e.g., ELISA validation, inter-/intra-assay variability for cortisol RIA) are needed.

    • The classification of disease severity (e.g., "mild," "moderate") lacks objective criteria (e.g., clinical scoring system), introducing potential subjectivity.

  3. Results:

    • Data presentation is thorough, but some figures lack clarity. For instance, Figure 1’s y-axis labels for haptoglobin (units: ng/mL vs. mg/mL?) and inconsistent scaling between ferritin and IL1β/IL6 in Figures 2–4 complicate interpretation.

    • The weak correlations reported (e.g., q=0.49 for ferritin-IL1β) raise questions about clinical significance. A discussion of effect sizes or biological relevance is missing.

    • Table 1 requires clearer column headings (e.g., "Grouping" should specify treatment/sex subgroups).

  4. Discussion:

    • The link between elevated cortisol in diseased calves and prior studies is well-addressed, but the unexpected lack of iron supplementation effects warrants deeper exploration (e.g., dose adequacy, timing).

    • Sex-specific findings (e.g., male-specific correlations) are highlighted but not mechanistically explained. Hypotheses for these differences (e.g., hormonal influences) should be proposed.

    • The clinical implications of diminished FRAP and elevated ferritin in inflammation are underdeveloped. How might these findings influence iron supplementation protocols or disease management?

  5. Figures and Tables:

    • Figures 1–7: Axes labels and units need standardization (e.g., "FRAP [μmol/L]" in Figure 7 vs. "Haptoglobin [ng/mL]" in Figure 1C). Error bars (SD vs. SEM?) should be clarified in captions.

    • Table 1: The term "Q" is used for Spearman coefficients but should be replaced with "r_s" or "ρ" to avoid confusion with false discovery rate (q-values).

Conclusion
The manuscript contributes novel data on iron metabolism and inflammation in neonatal calves, but revisions are necessary to address methodological transparency, statistical rigor, and clinical context. Strengthening the discussion of limitations (e.g., short observation period, farm-source variability) and providing actionable recommendations for future research (e.g., longitudinal studies, mechanistic investigations) would enhance impact. With minor grammatical edits and improved data presentation, this work will be suitable for publication.

Recommendations for Revision

  1. Simplify complex sentences and ensure consistent tense usage.

  2. Justify sample size and clarify disease severity classification criteria.

  3. Standardize figure axes, labels, and error bars; expand table/figure captions.

  4. Discuss the clinical relevance of weak correlations and sex-specific findings.

  5. Include quality control details for assays.

  6. Propose mechanistic hypotheses for observed associations.

This study holds promise for informing veterinary practices but requires refinement to maximize its contribution to the field.

Comments on the Quality of English Language

The study investigates the interplay between iron metabolism, stress, and inflammatory markers in neonatal calves, addressing a relevant topic in veterinary physiology. While the research design is methodical and the data provide valuable insights, several areas require improvement to enhance clarity, statistical robustness, and clinical relevance.

Grammar and Syntax
The manuscript is generally well-written, but some sentences are overly complex or lengthy, reducing readability. For example:

  • "Neither iron supplementation, disease status, nor sex had statistically significant effects on the areas under the curve of ferritin, WBC, TP, IL1β, IL6, SP, hepcidin, haptoglobin, or FRAP."
    This sentence could be split for clarity. Minor grammatical errors (e.g., inconsistent tense usage in the Results section) and punctuation issues (e.g., missing commas in lists) are present but do not severely impede comprehension.

Structure and Content

  1. Introduction: Adequately contextualizes the study, but the mechanistic link between stress, inflammation, and iron metabolism could be expanded. The rationale for focusing on cortisol and specific cytokines (e.g., IL1β, IL6) is clear, but a brief explanation of why these markers were prioritized over others would strengthen the section.

  2. Materials and Methods:

    • The sample size (n=40) may limit statistical power, particularly for subgroup analyses (e.g., sex-specific effects). Justification for the sample size or a post-hoc power analysis is lacking.

    • Detailed descriptions of quality control measures for assays (e.g., ELISA validation, inter-/intra-assay variability for cortisol RIA) are needed.

    • The classification of disease severity (e.g., "mild," "moderate") lacks objective criteria (e.g., clinical scoring system), introducing potential subjectivity.

  3. Results:

    • Data presentation is thorough, but some figures lack clarity. For instance, Figure 1’s y-axis labels for haptoglobin (units: ng/mL vs. mg/mL?) and inconsistent scaling between ferritin and IL1β/IL6 in Figures 2–4 complicate interpretation.

    • The weak correlations reported (e.g., q=0.49 for ferritin-IL1β) raise questions about clinical significance. A discussion of effect sizes or biological relevance is missing.

    • Table 1 requires clearer column headings (e.g., "Grouping" should specify treatment/sex subgroups).

  4. Discussion:

    • The link between elevated cortisol in diseased calves and prior studies is well-addressed, but the unexpected lack of iron supplementation effects warrants deeper exploration (e.g., dose adequacy, timing).

    • Sex-specific findings (e.g., male-specific correlations) are highlighted but not mechanistically explained. Hypotheses for these differences (e.g., hormonal influences) should be proposed.

    • The clinical implications of diminished FRAP and elevated ferritin in inflammation are underdeveloped. How might these findings influence iron supplementation protocols or disease management?

  5. Figures and Tables:

    • Figures 1–7: Axes labels and units need standardization (e.g., "FRAP [μmol/L]" in Figure 7 vs. "Haptoglobin [ng/mL]" in Figure 1C). Error bars (SD vs. SEM?) should be clarified in captions.

    • Table 1: The term "Q" is used for Spearman coefficients but should be replaced with "r_s" or "ρ" to avoid confusion with false discovery rate (q-values).

Conclusion
The manuscript contributes novel data on iron metabolism and inflammation in neonatal calves, but revisions are necessary to address methodological transparency, statistical rigor, and clinical context. Strengthening the discussion of limitations (e.g., short observation period, farm-source variability) and providing actionable recommendations for future research (e.g., longitudinal studies, mechanistic investigations) would enhance impact. With minor grammatical edits and improved data presentation, this work will be suitable for publication.

Recommendations for Revision

  1. Simplify complex sentences and ensure consistent tense usage.

  2. Justify sample size and clarify disease severity classification criteria.

  3. Standardize figure axes, labels, and error bars; expand table/figure captions.

  4. Discuss the clinical relevance of weak correlations and sex-specific findings.

  5. Include quality control details for assays.

  6. Propose mechanistic hypotheses for observed associations.

This study holds promise for informing veterinary practices but requires refinement to maximize its contribution to the field.

Author Response

Dear Editor and Reviewers,

The authors would like to thank you for the evaluation of our manuscript, and your time and efforts to provide constructive comments and remnant concerns. You have pointed out important aspects, which have been taken into careful consideration. The authors hope that the edits made are to your satisfaction and that the amended manuscript will be considered for publication in Animals.

“Your manuscript has now been reviewed by experts in the field and can be found with the review reports at: https://susy.mdpi.com/user/manuscripts/resubmit/716cb0e5e61ad56c1b951fbccff627b7

Please revise the manuscript found at the above link according to the reviewers' comments and upload the revised file within 10 days.”

The authors have addressed each of the comments below.

Reviewer 3

Open Review

Quality of English Language

(x) The English could be improved to more clearly express the research.
( ) The English is fine and does not require any improvement.

Yes

Can be improved

Must be improved

Not applicable

Does the introduction provide sufficient background and include all relevant references?

(x)

( )

( )

( )

Is the research design appropriate?

( )

(x)

( )

( )

Are the methods adequately described?

( )

(x)

( )

( )

Are the results clearly presented?

(x)

( )

( )

( )

Are the conclusions supported by the results?

(x)

( )

( )

( )

Comments and Suggestions for Authors

The study investigates the interplay between iron metabolism, stress, and inflammatory markers in neonatal calves, addressing a relevant topic in veterinary physiology. While the research design is methodical and the data provide valuable insights, several areas require improvement to enhance clarity, statistical robustness, and clinical relevance.

Answer:

Thank you very much for your time and effort in reading our manuscript and giving this constructive feedback. We greatly appreciate your criticism and are sure that your advice will improve the quality and readability of our paper. In the following sections we will address each point of the reviewer’s suggestions and critical comments.

Grammar and Syntax
The manuscript is generally well-written, but some sentences are overly complex or lengthy, reducing readability. For example:

  • "Neither iron supplementation, disease status, nor sex had statistically significant effects on the areas under the curve of ferritin, WBC, TP, IL1β, IL6, SP, hepcidin, haptoglobin, or FRAP."
    This sentence could be split for clarity. Minor grammatical errors (e.g., inconsistent tense usage in the Results section) and punctuation issues (e.g., missing commas in lists) are present but do not severely impede comprehension.

Answer:

We deeply regret that the manuscript contains grammatical errors and overly complex or lengthy sentences. Our manuscript draft had been edited professionally by editage.com and we will send it back for reworking on the reviewed version. We will explicitly ask the managing editor to concentrate on this point.

Structure and Content

  1. Introduction: Adequately contextualizes the study, but the mechanistic link between stress, inflammation, and iron metabolism could be expanded. The rationale for focusing on cortisol and specific cytokines (e.g., IL1β, IL6) is clear, but a brief explanation of why these markers were prioritized over others would strengthen the section.

Answer:

To briefly explain the rationale behind choosing the specific cytokines, we have included an explanation on how those cytokines influence the serum concentrations of iron and ferritin. We hope that the included passage (lines 89–96) meets the point of criticism and satisfies the reviewer’s concerns.

  1. Materials and Methods:
    • The sample size (n=40) may limit statistical power, particularly for subgroup analyses (e.g., sex-specific effects). Justification for the sample size or a post-hoc power analysis is lacking.

Answer:

The sample size was based on a power analysis that had been calculated by the Department for Biomathematics and Data Processing of the University of Giessen. We have included this information in the Statistics and Data Analyses section (lines 222–224).

  • Detailed descriptions of quality control measures for assays (e.g., ELISA validation, inter-/intra-assay variability for cortisol RIA) are needed.

Answer:

Our examinations used commercially available ELISA test kits. All examinations were performed in double settings, and standard calibration curves were used. The quality control measures for the tests have been added (lines 177–181) and the same for the cortisol RIA were already given in the manuscript in lines 191–192 as follows: “Intra- and inter-assay coefficients of variation were 4.3% and 9.8%, respectively, with a detection limit of 2 nmol/L.”

  • The classification of disease severity (e.g., "mild," "moderate") lacks objective criteria (e.g., clinical scoring system), introducing potential subjectivity.

Answer:

Thank you for your comment. The severity of the disease was determined based on the duration of clinical signs and by using a scoring system that indicated an alteration in their vital parameters and signs of disease (diarrhea, omphalitis, or thrombophlebitis). This information is provided in the “Evaluation of Health Status” section in lines 198–205. Scoring criteria may be provided as supplemental material if the reviewer would prefer to have this background information available for readers.

  1. Results:
    • Data presentation is thorough, but some figures lack clarity. For instance, Figure 1’s y-axis labels for haptoglobin (units: ng/mL vs. mg/mL?) and inconsistent scaling between ferritin and IL1β/IL6 in Figures 2–4 complicate interpretation.

Answer:

We are terribly apologetic, but the rationale for this criticism is not clear to us. The presented data is given within the figures to visualize the timely courses of the examined parameters. The units of the single parameters are not relevant to compare congruent or incongruent developments over time. We have used conventional units for all parameters, and in our opinion, the figures are not inconsistently scaled. I am afraid that we may miss the reviewer’s point of criticism, and would appreciate further clarification here.

  • The weak correlations reported (e.g., q=0.49 for ferritin-IL1β) raise questions about clinical significance. A discussion of effect sizes or biological relevance is missing.

Answer:

We concede to the reviewer’s opinion that the rather weak correlations reported might lack biological relevance. However, the consistent correlations between ferritin and inflammatory markers in our study are supposed to point towards the necessity to critically overthink iron supplementation as a standard treatment in neonatal calves. To emphasize this, we have included the following passage in the manuscript: “In comparison to other studies [1-3, 7], iron-dosage, route of administration, and time of treatment may have also influenced the results. Our study used a rather low dosage of iron dextran of 10 mg/kg of iron dextran to mimic often used standard procedures in the field. Application of higher iron dosages or repeated treatments might have resulted in stronger correlations between the examined parameters. Meanwhile, biological relevance of existing correlations and their underlying effect sizes have to be taken into account. Despite being weak, the consistent correlations between ferritin and markers of inflammation presented in our study are suggested to encourage veterinarians in the field to critically overthink iron supplementations as a standard treatment procedure in neo-natal calves.” (lines 559–568).

  • Table 1 requires clearer column headings (e.g., "Grouping" should specify treatment/sex subgroups).

Answer:

To make the column heading clearer, we have changed the wording to “Grouping by sex or treatment” (see Table 1 lines 299 ff).

  1. Discussion:
    • The link between elevated cortisol in diseased calves and prior studies is well-addressed, but the unexpected lack of iron supplementation effects warrants deeper exploration (e.g., dose adequacy, timing).

Answer:

Iron supplementation slightly reduced cortisol concentrations in the calves of our study. However, these variances were not statistically significant. We have tried to address the reviewer`s criticism by including the following possible explanations for not reaching significance at this point: “Significance might have been reached using higher dosages of iron or a prolonged oral supplementation of iron and selenium as has been shown in human athletes [40] and young bulls transported over long-distances [41]. In athletes, a reduction of cortisol was achieved after oral supplementation of iron over a period of 3 weeks. In bulls, the application of slow-release selenium boluses resulted in an elevation of serum iron and a reduction of cortisol. The link between cortisol and iron metabolism is regarded to be an indirect effect of a promotion of inflammation in acute stress (i.e. higher cortisol concentrations in diseased animals). In animals suffering from acute stress, proinflammatory cytokines are enhanced leading to higher concentrations of hepcidin [6], thereby reducing iron absorption. Prolonged application of low-dose iron has been shown to reduce cortisol [40], but the correlating hormonal feedback regulation mechanisms have not been examined in detail. However, the present study concentrated on the effects of an iron injection as a single dosage often used in the field. The elevated cortisol levels in the animals of the present study result from acute stress due to birth, transportation, and disease. It is not predictable if higher single iron dosages or repeated application of iron might have altered the results.” (lines 430–445)

  • Sex-specific findings (e.g., male-specific correlations) are highlighted but not mechanistically explained. Hypotheses for these differences (e.g., hormonal influences) should be proposed.

Answer:

Concerning sex specific findings, we have included the following paragraph in the discussion section to give a general rationale as to why sex might influence the chosen parameters: “Sex specific findings of the present study are regarded to be the result of hormonal in-fluences on the examined parameters. This is in line with negative correlations of serum ferritin with testosterone in boys [34], and estrogen being involved in the ex-pression of ferroportin [35], thus, influencing iron metabolism. Whereas several studies in human medicine emphasize the influence of testosterone and estrogen on iron metabolism, serum ferritin levels, and anti- or proinflammatory cytokines, corresponding data for veterinary medicine is scarce. Nevertheless, sex-specific differences for substance P in calves for example have been shown, which emphasizes the necessity for sex-specific considerations [36]. Recent studies in mice and humans indicate that sex chromosomes and gonadal hormones influence the number and functions of immune cells, the cytokine signaling pathways, and humoral responses [37]. Due to a differing expression of X-chromosomal and autosomal genes, and the abundant presence of sex steroid hormone receptors in many cell types, sex influences nearly all biological systems [37]. The future establishment of specific reference ranges and cut-off values according to sex are therefore generally recommended.” (lines 406–420).

  • The clinical implications of diminished FRAP and elevated ferritin in inflammation are underdeveloped. How might these findings influence iron supplementation protocols or disease management?

Answer:

We thank the reviewer for advising us to give a deeper insight into the oxidative status of animals. Our results suggest that farmers and veterinarians in the fields should re-evaluate the necessity of an iron supplementation in diseased calves. Due to the reduced anti-oxidative capacity, the animals show a reduced capacity to disarm iron overload thus, resulting in potential imminent tissue and enzyme damage via the production of ROS. To include this critical point in the discussion section, we have added the following wording: “Our findings of a reduced antioxidant capacity are also in line with Fu et al. [63], who could show a reduced total antioxidant capacity in diarrheic calves. However, it is not clear whether the altered antioxidative status is the cause or the result of diarrhea. Nonetheless, the results of the present studies suggest and explain the contraindication of iron supplementation in inflammatory diseases (i.e. diarrhea). Cytokine concentrations and enzyme activities are altered in ailing calves, thus, representing a reduced capacity to disarm iron overload which may subsequently cause tissue and enzyme damage via the production of ROS. Therefore, general iron supplementation treatments in calves with inflammatory diseases should be rejected.” (lines 528–536)

  1. Figures and Tables:
    • Figures 1–7: Axes labels and units need standardization (e.g., "FRAP [μmol/L]" in Figure 7 vs. "Haptoglobin [ng/mL]" in Figure 1C). Error bars (SD vs. SEM?) should be clarified in captions.

Answer:

We deeply regret the perceived lack of standardization in data presentation in our figures. However, we have provided the generally used units for each examined parameter. For FRAP (ferric reducing ability of plasma) determination, the results are generally given as µmol/L and for haptoglobin in mg/mL. The figure captions already state that, “Dots and squares indicate arithmetic means, with whiskers representing standard deviations.” Therefore, we would greatly appreciate if the reviewer could accept the chosen presentation, or would give us detailed advice on how to change the settings. However, we would also kindly ask the reviewer to take into consideration that the comparisons may be drawn without a relevance of the given units.

  • Table 1: The term "Q" is used for Spearman coefficients but should be replaced with "r_s" or "ρ" to avoid confusion with false discovery rate (q-values).

Answer:

We must admit that we are extremely confused concerning this point. We are unable to find the term “Q” within table 1 and are therefore, not able to make any changes here.

Conclusion
The manuscript contributes novel data on iron metabolism and inflammation in neonatal calves, but revisions are necessary to address methodological transparency, statistical rigor, and clinical context. Strengthening the discussion of limitations (e.g., short observation period, farm-source variability) and providing actionable recommendations for future research (e.g., longitudinal studies, mechanistic investigations) would enhance impact. With minor grammatical edits and improved data presentation, this work will be suitable for publication.

Answer:

We thank the reviewer very much for the positive feedback, and would like to express our appreciation of the work and time invested in the review of our study. We have attempted to address all the comments and would be glad if our efforts could meet the reviewer’s concerns.

Recommendations for Revision

  1. Simplify complex sentences and ensure consistent tense usage.
  • We have asked our commercial editing service to address these points and are very confident that the wording could be further improved.
  1. Justify sample size and clarify disease severity classification criteria.
  • We have included the details of the power analysis that had been performed prior to the study. Classification scheme or scoring scheme of the study may be made available if the reviewer would request us to do so.
  1. Standardize figure axes, labels, and error bars; expand table/figure captions.
  • Please kindly refer to the points above and our rationale on why we would like not to change the figures concerning the units. The meaning of the error bars has already been given in each figure caption.
  1. Discuss the clinical relevance of weak correlations and sex-specific findings.
  • We have extended the discussion section to include clinical relevance of weak correlations and sex-specific findings.
  1. Include quality control details for assays.
  • The quality control details have been added in the material and methods section.
  1. Propose mechanistic hypotheses for observed associations.
  • We have strived to meet the reviewer’s points of criticism and have proposed possible underlying mechanisms that might support our findings.

This study holds promise for informing veterinary practices but requires refinement to maximize its contribution to the field.

Answer:

We greatly appreciate the reviewer’s positive opinion concerning our work and study. We have thoroughly edited the manuscript according to the suggestions made and hope that we could refine the paper draft to the reviewer’s satisfaction. We would like to thank the reviewer again for acknowledging our attempts to fulfill all recommendations and to address critical remarks. The reviewer’s work highly contributed to an overall improvement of our manuscript draft.

Submission Date

06 March 2025

Date of this review

13 Mar 2025 12:08:15

Round 2

Reviewer 1 Report

Comments and Suggestions for Authors I believe that many of the points that required attention were fixed, which gave the article much more quality.

Reviewer 2 Report

Comments and Suggestions for Authors To the Authors:
We extend our sincere gratitude for your meticulous responses to the review comments and the rigorous revisions implemented in the manuscript. The revised version demonstrates substantial improvements in methodological rigor, depth of mechanistic exploration, and comparative analysis with existing literature. Notably, the supplementation of validation data for detection methods (e.g., intra-/inter-assay coefficients of variation), the elucidation of hepcidin/ROS-mediated mechanisms, and the comparative alignment with findings from Panousis et al. (2018) have significantly enhanced the scientific value of the study. We confirm that the manuscript now adequately addresses the concerns raised during the peer review process.